# Vegetation and Precipitation Patterns Define Annual Dynamics of CO₂ Efflux from Soil and Its Components

Dmitriy Khoroshaev [1,*] , Irina Kurganova [1], Valentin Lopes de Gerenyu [1], Dmitry Sapronov [1], Sergey Kivalov [1] , Abeer S. Aloufi [2] and Yakov Kuzyakov [3,4]

[1] Institute of Physicochemical and Biological Problems in Soil Science of the Russian Academy of Sciences, 142290 Pushchino, Russia; kurganova@pbcras.ru (I.K.); lopes@pbcras.ru (V.L.d.G.); sapronov@pbcras.ru (D.S.); kivalov@pbcras.ru (S.K.)

[2] Department of Biology, College of Science, Princess Nourah Bint Abdulrahman University, P.O. Box 84428, Riyadh 11671, Saudi Arabia; asaloufi@pnu.edu.sa

[3] Agro-Technological Institute, Peoples Friendship University of Russia (RUDN University), 117198 Moscow, Russia; kuzyakov@gwdg.de

[4] Department of Soil Science of Temperate Ecosystems, Georg-August University of Göttingen, 37077 Göttingen, Germany

* Correspondence: d.khoroshaev@pbcras.ru

**Abstract:** Respiration of soil heterotrophs—mainly of bacteria and fungi—is a substantial part of carbon balance in terrestrial ecosystems, which tie up organic matter decomposition with the rise of atmospheric CO₂ concentration. Deep understanding and prediction of seasonal and interannual variation of heterotrophic and autotrophic components of CO₂ efflux from soil is limited by the lack of long-term, full-year measurements. To better understand the impact of current climate changes on CO₂ emissions from soils in the mixed forest and mowed grassland, we measured CO₂ efflux every week for 2 years. Heterotrophic (SOM-derived + leaf litter) and root-associated (root with rhizosphere microorganisms) components were partitioned by the root exclusion method. The total CO₂ efflux from soil was averaged 500 g C m⁻² yr⁻¹ in the forest and 650 g C m⁻² yr⁻¹ in the grassland, with shares of the no-growing cold season (Nov–Mar) of 22% and 14%, respectively. The heterotrophic component of CO₂ efflux from the soil averaged 62% in the forest and 28% in the grassland, and it was generally stable across seasons. The redistribution of the annual precipitation amounts as well as their deficit (droughts) reduced soil respiration by 33–81% and heterotrophic respiration by 24–57% during dry periods. This effect was more pronounced in the grassland (with an average decline of 56% compared to 39% in the forest), which is related to lower soil moisture content in the grassland topsoil during dry periods.

**Keywords:** microbial respiration; root derived respiration; grassland and forest ecosystems; seasonal dynamics; interannual variability

## 1. Introduction

The CO₂ efflux from soils, frequently termed as soil respiration (SR) [1,2], is the second-largest flux between the Earth's surface and atmosphere after photosynthesis [3,4]. SR reflects mainly biological processes, but it includes two distinct components: respiration of plant roots (RR) and heterotrophic microorganisms in soil (HR). The first, with above-ground plant biomass respiration, constitutes losses of gross primary production (GPP), and the second reflects the decomposition of organic matter entering and available in soil [5]. Therefore, only part of SR is linked with a pathway of carbon loss from soil. The specific functional role of these CO₂ sources in the ecosystem carbon cycle makes it necessary to separate the SR into root and heterotrophic components to relate soil carbon turnover with global warming. There are concerns that an increase in global temperature may increase the ratio of HR to net primary production (NPP = GPP − plant respiration). Hence, increasing

HR vs. NPP due to rising temperatures may reduce the ability of ecosystems to absorb $CO_2$ released through anthropogenic activities due to the fertilization effect [6].

The proximity of global estimates of NPP in terrestrial ecosystems (56 Gt C/year) [7] and HR (51 Gt C/year) [8–11] imposes special requirements to reduce their uncertainty. However, all methods are characterized by a high degree of uncertainty in the estimates, their spatial distribution and variability, as well as temporal dynamics, whether they involve extrapolating SR and its components from local measurements to regional and global scales (bottom-up approach) or estimating them using GPP and NPP data (top-down approach) [4,8,10,12–14]. Despite the growth of observational data and the application of machine learning techniques for their analysis, there has not been a convergence in global estimates of SR. Instead, there has been an increase in the variability of these estimates [13]. Since HR is often calculated based on an empirical relationship with SR or obtained as a product of Earth system models [10], or scaled based on a smaller HR dataset compared to SR [15], it is clear that there is a problem with the scaling of HR, similar to the scaling of SR.

The type of ecosystem and vegetation cover have a strong effect on SR because they are responsible for the input of organic compounds into soil, microclimate, biodiversity, and food webs, as well as soil properties. Reflecting the total biological activity of an ecosystem, SR and, particularly, HR are controlled both by the climatic conditions and vegetation type [16–18]. The average contribution of root respiration to the total SR flux from the soils varies from 38% in agrocenoses to 63–72% in the tundra and northern taiga ecosystems [1]. However, the contribution of RR to SR can vary significantly depending on the time, place, and method of $CO_2$ flow separation, with values ranging from less than 10% to more than 90% [19,20]. The seasonal and interannual soil $CO_2$ fluxes and their structure are linked with the ecosystem type [21–23], whereas the reaction of the SR to the climatic changes and extremal events [24] may be associated with the individual soil carbon pools with different turnover times [25]. The importance of regularly monitoring the SR structure is highlighted by the presence of seasonal dynamics specific to different ecosystems [22,26,27]. The heterogeneous response of separate SOM components to changing climatic conditions deepens our understanding of the relationship between global warming and the terrestrial carbon cycle.

We estimated the seasonal dynamics of SOM-derived and root-derived components of SR in forest and grassland ecosystems for two consecutive years. Our objectives were (i) to compare ecosystem-driving seasonal dynamics of HR and RR; and (ii) to evaluate the response of these components to changing weather conditions. In starting this monitoring, we assumed that the contrast between plant life forms on the same soils allows us to reveal the features of seasonal variation in the SR structure in various ecosystem types. Additionally, we expected clear seasonal dynamics of the observed components, based on previous estimates of seasonal dynamics of $CO_2$ sources from the soil [22]. Given that grassland ecosystems are enriched with thin, physiologically active roots compared to forests, we hypothesized (i) that the main difference in yearly SR dynamics between these two contrasting ecosystems is associated with RR dynamics. Considering the individual microclimates and adaptation strategies of grassland and forest ecosystems to weather anomalies, we hypothesized (ii) that there would be various magnitudes of HR and RR responses to exceptional weather conditions.

## 2. Materials and Methods

### 2.1. Site Description

Study sites are located within the Priuksko-Terrasny Nature Biosphere Reserve (Moscow region, Russia) near the integrated background monitoring of environmental pollution station (IBMon; 54°54.148′ N, 37°33.377′ E). The first site (Figure 1a) was located in the native mixed forest (>130 years; *Pinus sylvestris*, *Tilia cordata*, *Populus tremula*, *Picea abies*, *Acer platanoides*, *Betula* spp.). The forest stand had formed during natural reforestation after pine logging in the 1930s [28]. The second site (Figure 1b) was located at a distance of

100 m on the moving grassland (>50 years developed) with a polydominant association (*Alchemilla* sp., *Viola tricolor* L., *Trisetum flavescens* (L.) Beauv., *Rumex acetose* L.).

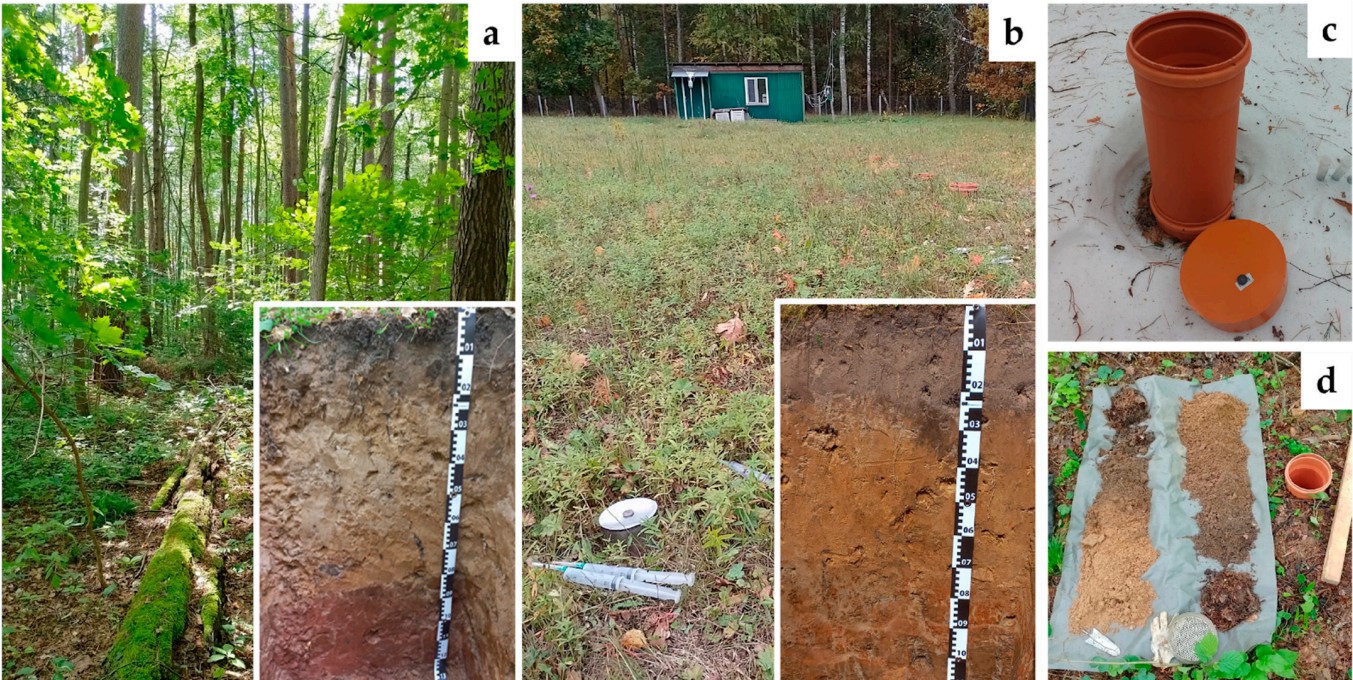

**Figure 1.** General view of the vegetation and soils within the native mixed forest (**a**) and mowed grassland (**b**) ecosystems, along with chambers for soil respiration measurements during warm (**b**) and cold (**c**) periods. The process of installing soil chambers containing root-free soils (**d**).

The soil was classified as a weak differential Entic Podzol (Arenic) [29] on sandy alluvial-glaciofluvial deposits of the Oka glaciofluvial alluvial plain (Figure 1a,b). The soil texture of topsoil (0~10 cm) is sandy loam with clay-silt content (<0.01 mm) of 8.4–9.4% and sand for the deeper soil layer (10~30 cm). The thickness of litter horizon in the forest is approximately 2–3 cm, and litter storage amounts to $550 \pm 120$ g C m$^{-2}$. In contrast, the litter horizon in grassland is shallow and sparse, with a storage of approximately $100 \pm 17$ g C m$^{-2}$. Soil under grassland has a well-developed sod horizon (Ah) at a depth of 0–10 cm. The soils of the two ecosystems showed no marked differences in their properties (Table 1).

**Table 1.** Physicochemical soil properties of the monitoring plots in the forest and grassland.

| Place | Depth cm | pH | Bulk Density g cm$^{-3}$ | $C_{org}$ g kg$^{-1}$ | $N_{tot}$ g kg$^{-1}$ | C/N | $C_{mic}$ mg kg$^{-1}$ |
|---|---|---|---|---|---|---|---|
| Forest | 0–5 | $5.03 \pm 0.12$ a | $0.93 \pm 0.05$ a | $35.13 \pm 3.42$ a | $2.23 \pm 0.15$ a | $15.6 \pm 0.6$ a | $150 \pm 35$ ab |
| | 5–10 | $4.15 \pm 0.12$ bc | $1.39 \pm 0.06$ b | $16.59 \pm 0.94$ c | $1.20 \pm 0.10$ b | $13.9 \pm 0.8$ a | $85 \pm 11$ bc |
| | 10–20 | $4.01 \pm 0.11$ c | $1.32 \pm 0.05$ b | $7.85 \pm 0.84$ de | $0.55 \pm 0.05$ cd | $14.2 \pm 0.9$ a | $42 \pm 9$ c |
| | 20–30 | $4.06 \pm 0.15$ c | $1.50 \pm 0.08$ b | $4.00 \pm 0.58$ e | $0.30 \pm 0.04$ d | $13.4 \pm 0.2$ a | $34 \pm 4$ c |
| Grassland | 0–5 | $4.51 \pm 0.03$ b | $0.93 \pm 0.07$ a | $25.34 \pm 1.25$ b | $2.34 \pm 0.19$ a | $10.9 \pm 0.5$ b | $211 \pm 11$ a |
| | 5–10 | $3.97 \pm 0.05$ c | $1.39 \pm 0.12$ b | $11.64 \pm 0.28$ cd | $1.21 \pm 0.04$ b | $9.6 \pm 0.3$ b | $83 \pm 2$ bc |
| | 10–20 | $4.00 \pm 0.04$ c | $1.53 \pm 0.04$ b | $8.01 \pm 0.57$ de | $0.84 \pm 0.06$ bc | $9.5 \pm 0.1$ b | $43 \pm 1$ c |
| | 20–30 | $4.10 \pm 0.05$ bc | $1.56 \pm 0.06$ b | $4.00 \pm 0.41$ e | $0.44 \pm 0.04$ cd | $9.0 \pm 0.2$ b | $31 \pm 2$ c |

$\pm$standard error (n = 3–4). pH was measured in KCl (1 M). Organic carbon ($C_{org}$) and total nitrogen ($N_{tot}$) contents were determined by the dry combustion method (LECO-932, St. Joseph, MI, USA). Microbial carbon ($C_{mic}$) was analyzed by the substrate-induced respiration method [30]. Different letters indicate pairs of average values, the differences of which are detected during the multiple comparison procedure (Tukey test, $\alpha$ = 5%) after two-way ANOVA (Ecosystem × Depth).

## 2.2. Climate

The climate of the region is temperate continental (Dfb according to the Köppen climate classification). Average annual temperature during 1991–2020 (with standard deviation) provided by IBMon for the study region is 5.7 (0.8) °C, and annual amount of precipitation is 640 (107) mm. Approximately 1/3 of precipitation falls over summer (June–August) and 2/3 during the warm season (April–October). The annual average temperatures of July and January over this period were 18.8 (1.9) and −7.2 (3.2) °C, respectively. The permanent snow cover in this region typically lasts from November–December to March–April. The average duration of snow occurrence is 131 (23) days with an average maximum height of 53 (12) cm.

## 2.3. Soil Respiration Measurement

The nonsteady-state, nonthrough-flow close chamber method (in classification [31]) was used to measure the rate of $CO_2$ efflux from soil. During the nonsnow period, 5 steel chambers with a diameter of 10 cm and a height of 12 cm were used for SR measurements (Figure 1b). The top of the chamber was equipped with a rubber septum for collecting gas samples. These chambers were inserted into the soil at a depth of 4 cm after cutting of green plant parts until 1 cm height. Gas samples were collected using 23 mL plastic syringes at 0, 10, and 20 min after the installation of the chambers.

During the snow period, 5 chambers were PVC plastic pipes with a diameter of 20 cm and a height of 60 cm, which were stationary inserted into the soil at a depth of 10 cm after removing a living plant's canopy (Figure 1c). These were installed shortly before the first frost and removed after the melting of the snow cover. The chambers were supplied with additional sections when the snow depth exceeded the chambers height. Gas samples were collected using syringes at 0, 20, and 40 min after closing the chambers with cups. Gas samples had been immediately moved to a laboratory and analyzed with an infrared $CO_2$ analyzer (Li-Cor LI-820, Lincoln, NE, USA)

## 2.4. Partitioning of Total $CO_2$ Efflux from Soil for Contribution of Heterotrophs and Root-Derived $CO_2$

HR was estimated by the root exclusion method. PVC plastic pipes of 20 cm diameter were cut into the soil at the depth of 45 cm. The upper part of the pipes protruded from the soil by 10–15 cm and was chamber walls that closed with cups with rubber septa. During the cutting process, soil layers of 2-cm thickness were removed step by step from the inner space of the pipes (Figure 1d). The soil had been freed from roots and was then sieved through a 5-mm screen. Prepared soil was removed into the pipes while maintaining the order of layers and density (compacting soil by a wooden pole) after the installation of the pipes at the indicated depth. It was possible to install the pipes without severely disturbing its structure due to the sandy structure of the soil. Leaf litter horizon was preserved in the forest site. Due to the absence of litter horizon in the grassland site, the surface of the bare soil in the chambers was covered by the nonwoven material that fits into the diameter of the pipe. This prevents the disturbance of soil surface during rain and reduces overheating and over-drying of the topsoil. HR measurements were performed similarly to soil respiration.

We prepared and installed 5 chambers for HR measurements at the end of May 2022. The measurements of $CO_2$ emissions began 1 month after the installation of HR chambers. A posteriori comparison of $CO_2$ efflux from HR chambers installed in different years allowed us to establish that a period of 2–3 months is sufficient for the stabilization of $CO_2$ efflux from the soil, but a major decrease in extra flux is observed during the first month (Figure A1). There was no visible systematic decrease in HR observed over the two years based on a comparison of two series of chambers that were installed in consecutive years. The soil in the chambers was somewhat wetter than that around it due to the absence of plant transpiration (Figure A2).

The above-described method allows us to measure HR associated with the decomposition of SOM as well as litter in the forest. It is worth considering that when soil was

sieved, the part detritus fraction (up to 5 mm) and the remains of thin roots remained in the soil. Note that the pool of soil detritus in the topsoil of HR chambers in the forest was replenished by leaf and stem litter, whereas the replenishment of aboveground plant residues in the grassland was absent. The RR included the $CO_2$ from roots and rhizosphere microbiomes.

*2.5. Soil Temperature and Moisture Measurement*

The air and soil temperatures were measured at a depth of 5 cm using a portable thermometer (Hanna Checktemp-1, Nusfalau, Romania; the measurement error was $\pm 0.3$ °C) directly at the measuring time of the SR rate. The moisture content of the soil in the 0–5 cm layer and forest litter was determined using the gravimetric method (drying at 105 °C for at least 6 h). The snow cover depth was also measured during the snow time period.

*2.6. Data Processing*

Rate of SR (mg C m$^{-2}$ h$^{-1}$) flux from soil was calculated by the Equation (1):

$$SR = \frac{dCO_2}{dt} \frac{(V_{ch} - V_{sn})M}{SV_m 100}, \tag{1}$$

where $dCO_2/dt$—rate of rise $CO_2$ into chamber, ppm h$^{-1}$; $V_{ch}$—chamber volume, cm$^3$, $V_{sn}$—snow volume, cm$^3$, $S$—chamber base area, cm$^2$; $M$—molar mass of carbon (12 g mol$^{-1}$), $V_m$—molar volume of gas at standard conditions (22.4 mol L$^{-1}$). Snow density data was taken from specialized arrays for climate research for the nearest weather station, Serpukhov [32].

The total SR flux (TSR, g C m$^{-2}$ period$^{-1}$) was calculated by linear interpolation of the measured values between successive dates in accordance with Equation (2):

$$TSR = \sum_{i=1}^{n} (\overline{R}_n t_n), \tag{2}$$

where $\overline{R}_n$—average rate of $CO_2$ flux between two consecutive dates (g C m$^{-2}$ day$^{-1}$), $t_n$—duration of the period between two consecutive dates (days).

Bootstrap analysis was used for estimation of seasonal SR, HR, and RR fluxes and their 95% confidence intervals (5000 iterations). We used the R base library (v.4.4.1); part of the measurements in June 2022 was calculated on the basis of a regression equation of HR from SR due to missed HR measurements in the first month after installing soil chambers (Figure A4). A standard error was used as a measure of variability for means. A standard significance level of $\alpha = 5\%$ has been accepted in general cases.

## 3. Results

*3.1. Weather Conditions During the Monitoring Period*

In general, the years 2022 and 2023 were characterized by similar mean annual temperature (MAT) and annual precipitation amount (PA) that did not differ from the reference period (1991–2020) but were characterized by important anomalies of weather conditions in certain seasons (Table 2).

The wet and cool spring of 2022 was followed by a dry summer (June–August) with the PA of 109 mm, which was half of the PA during the reference period. There were three dry periods of 9–13 days from the end of July to the beginning of September, alternating with two short-term wet periods (3–4 days each). These dry periods were accompanied by the positive temperature anomaly in August. Autumn temperature, with the exception of cool September, was close to the reference period. A low, stable snow cover was formed in mid-November. A 50-year record amount of precipitation (liquid equivalent, mm) fell in December. However, these precipitations did not lead to a sharp increase in a snow cover amount due to the thaws. The snow cover reached maximum height by mid-March 2023, after which it melted to the first days of April 2023.

**Table 2.** The average monthly temperatures and sum of precipitations at the data of Integrated background monitoring of environmental pollution (54°54.148′ N, 37°33.377′ E) for the last meteorological reference period and the years of the monitoring $CO_2$ fluxes.

| Period | January | February | March | April | May | June | July | August | September | October | November | December | Year |
|---|---|---|---|---|---|---|---|---|---|---|---|---|---|
| | | | | | Air temperature, °C | | | | | | | | |
| 1991– | −7.2 | −6.6 | −1.2 | 6.7 | 13.0 | 16.5 | 18.8 | 17.1 | 11.6 | 5.8 | −1.1 | −5.2 | 5.7 |
| 2020 | (3.2) | (4.1) | (3.0) | (1.8) | (2.2) | (1.7) | (1.9) | (1.4) | (1.5) | (1.6) | (3.0) | (3.8) | (0.8) |
| 2022 | −5.7 | **−1.9 ↑** | −2.0 | 5.4 | **9.8 ↓** | 17.2 | 18.9 | **19.6 ↑** | **9.0 ↓** | 6.5 | −0.8 | −4.5 | 6.0 |
| 2023 | −5.4 | −4.7 | 0.7 | **8.6↑** | 11.2 | 15.2 | 17.5 | 18.4 | **13.2 ↑** | 4.9 | −0.6 | −4.7 | 6.2 |
| 2024 | **−11.1 ↓** | −4.8 | 0.1 | **10.4↑** | −11.1 | nd | nd | nd | nd | nd | nd | nd | nd |
| | | | | | Precipitation amount, mm | | | | | | | | |
| 1991– | 44 | 39 | 36 | 39 | 56 | 75 | 82 | 63 | 57 | 61 | 44 | 44 | 640 |
| 2020 | (16) | (14) | (17) | (19) | (31) | (36) | (35) | (43) | (38) | (30) | (22) | (20) | (107) |
| 2022 | **82 ↑** | **19 ↓** | **13 ↓** | **98 ↑** | 75 | **34 ↓** | 54 | 22 | 87 | 73 | 37 | **125 ↑** | 718 |
| 2023 | **22 ↓** | 42 | **76 ↑** | 52 | **19 ↓** | 67 | 79 | 46 | **15 ↓** | **113 ↑** | **101 ↑** | **85 ↑** | 716 |
| 2024 | 55 | 43 | **7 ↓** | 51 | 36 | nd | nd | nd | nd | nd | nd | nd | nd |

Note: The standard deviation is shown in the brackets. The bold font with the signs ↑ or ↓ indicates meteorological parameters that deviate from the reference period by more than 1 standard deviation.

May 2023 was anomalously dry. Although the total summer PA was close to the reference period, its substantial part fell during short periods. In June, 52 out of 67 mm of precipitation fell on the first and last days of the month, leading to a 21-day continuous dry period. A similar pattern was observed in August, with most PA (38 mm) falling between the 20th and 26th of the month. An anomaly dry and warm September was followed by an anomaly wet October–December.

### 3.2. The Dynamics of $CO_2$ Efflux from Soil

Overall, mean annual SR rates were higher than HR by 1.7–1.9 times in the forest and by 4 times in the grassland (Figure A3). There are no marked interannual differences in variability and summary statistics of SR and HR (Table A1). The annual dynamics of SR and HR values reflected the changes in soil temperature. Short-term variations (less than one month) include both the periods of calm and marked waves of SR fluxes, reaching 150–200 mg C m$^{-2}$ h$^{-1}$ (Figure 2). In November and December 2022, anomalous growth of SR and HR values was observed after steady temperature transition from 0 and soil surface freezing at snow cover height < 10 cm. These anomalously high winter SR fluxes were excluded during calculations of total monthly and seasonal $CO_2$ fluxes.

The remarkable decline of SR rate was observed during the dry periods in August 2022, June 2023, and September 2023 (Figure 3). Soil moisture content was a substantial factor influencing SR dynamics during these periods (Figure 4). We estimated the effect size of a dry period as the difference between $CO_2$ fluxes on the first day of the dry period and the minimum $CO_2$ fluxes measured during that period. The decrease in $CO_2$ efflux during August 2022 was more pronounced compared to June 2023 and September 2023, due to a longer dry period (37 days vs. 21–22 days, respectively). The decline of SR and HR in the grassland during the dry periods was more than in the forest: 56% vs. 39% across all fluxes and periods (paired *t*-test, n = 6, *p* < 0.001). It was found that HR was less affected by dry periods compared to SR and, consequently, RR (Figure 3).

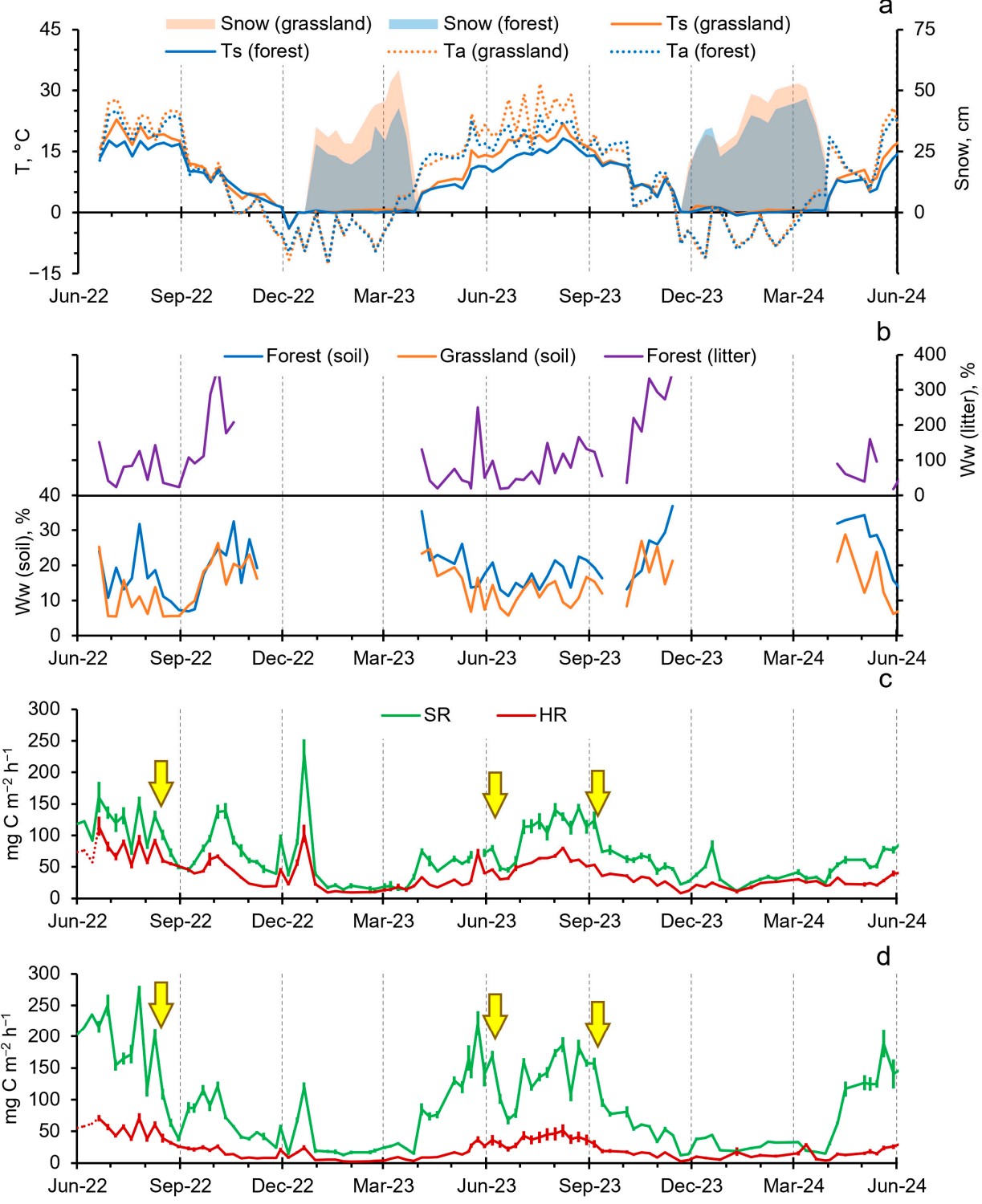

**Figure 2.** Dynamics of air (Ta) and soil (Ts) temperatures and height of snow cover (**a**), soil and forest litter moisture content (**b**) during the measurements of soil respiration (SR) and respiration of SOM−derived microorganisms (HR) in the forest (**c**) and the grassland (**d**). The dotted lines for HR show values reconstructed using the regression method for the first half of June 2022. Arrows indicate the decrease in SR and HR values during prolonged dry periods: August 2022, June 2023, and September 2023. Error bars are standard errors of the mean.

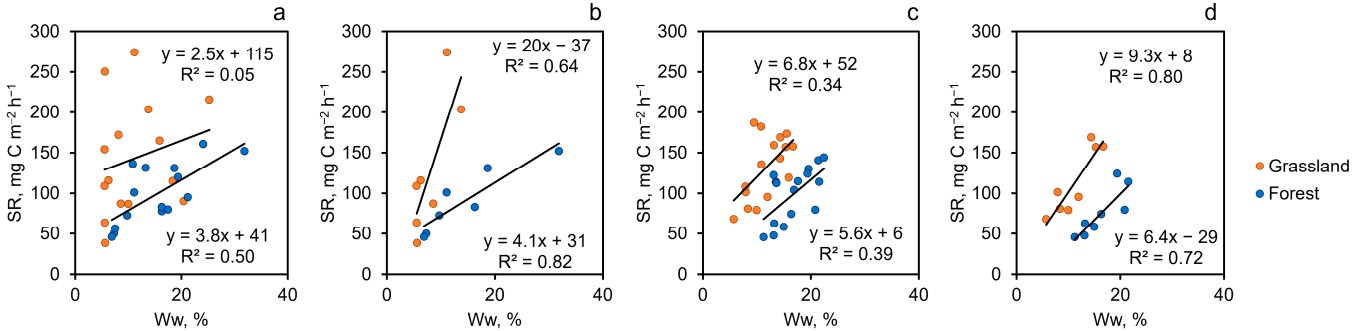

**Figure 3.** Dynamics of air (Ta) temperatures, day sum precipitation (P), soil moisture content (Ww) during the measurements of soil respiration (SR) and respiration of SOM-derived microorganisms (HR) in the grassland (**a**,**c**) and the forest (**b**,**d**) during summer–autumn periods of 2022 (**a**,**b**) and 2023 (**c**,**d**).

**Figure 4.** Soil respiration (SR) and soil moisture content (Ww) relationships during summer–autumn periods of 2022 (**a**,**b**) and 2023 (**c**,**d**) in the forest and the grassland. Data for the whole June–September period (**a**,**c**) and for periods with a high amount of dry days: 26 June–8 August 2022 (**b**) as well as June and 29 August–4 October 2023 (**d**).

### 3.3. Seasonal Fluxes of Total $CO_2$

There were clear seasonal variations in total monthly SR, HR, and RR values (Figure 5). Distinct differences were observed between total monthly HR and RR. Particularly, impressive growth of total monthly RR has been observed from March to May, in contrast to the smooth increase in HR values during this period. The decline of soil $CO_2$ fluxes observed during the summer dry periods has led to a reduction in total SR and RR in these months, but has relatively weakly affected HR values. We observed significant correlations between total monthly SR values (n = 24) in both ecosystems (r = 0.84) and between total monthly HR values as well (r = 0.96). For total monthly RR values, the relationship between ecosystems studied was weaker (r = 0.57).

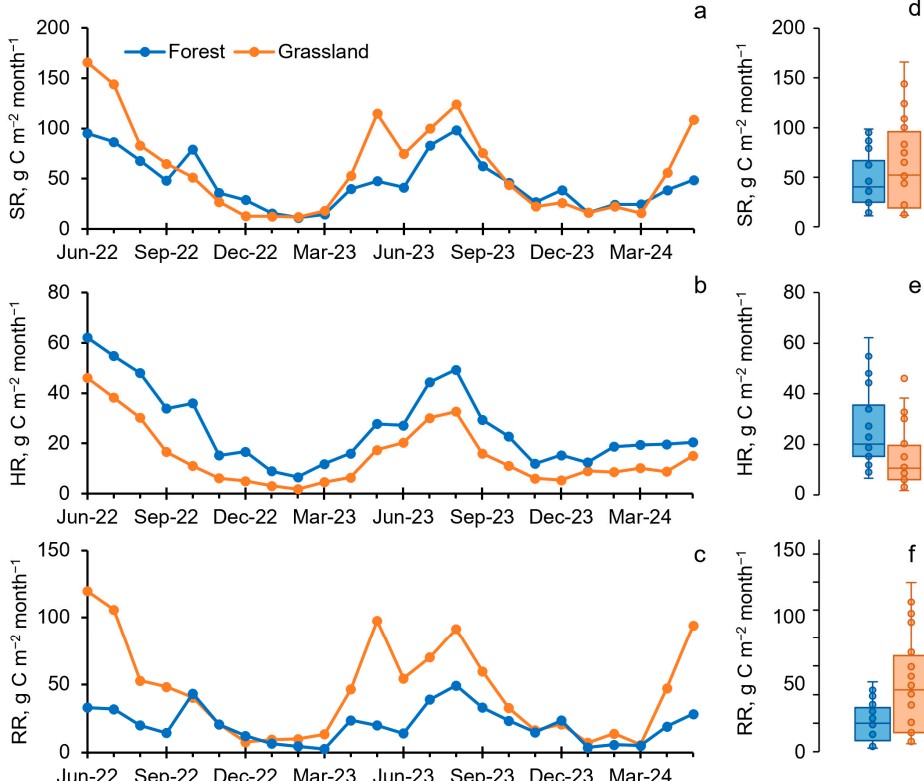

**Figure 5.** Dynamics of total monthly soil (SR), SOM-derived heterotrophs (HR), and root-derived (RR) respiration values (**a–c**) and their distribution over two years (**d–f**): the median (bar), lower (Q1) and upper (Q3) quartiles ("boxes"); X1 = Q1 − 1.5 IQR (interquartile range, IQR = Q3 − Q1) and X2 = Q3 − 1.5 IQR ("moustaches"); all data are shown as dots.

For the formal analysis, we divided the year into two seasons: October–March and April–September. Two-way ANOVA indicated a likely interaction between ecosystem type and the seasonal factor for SR ($p = 0.006$), along with a formally nonsignificant effect of the ecosystem at the annual scale ($p = 0.06$). For HR, the analysis revealed no interaction between the ecosystem and the season ($p = 0.59$), while a significant ecosystem effect was observed ($p = 0.0006$). All possible combinations of ecosystem and seasonal factors were significant for RR ($p \ll 0.001$). Therefore, seasonal differences in total month SR between the two ecosystems are caused mainly by the dynamics of RR flux (Figure 6a,c). The root component of total SR was notably higher in the grassland than in the forest between April and September, whereas it was similar in both ecosystems between October and March (Figure 5c). Alternate HR values were consistently higher in the forest than in the grassland during the entire year (Figure 5b). As with RR, the differences increased during the warm period, reflecting the response of HR to temperature increasing (Figure 6a).

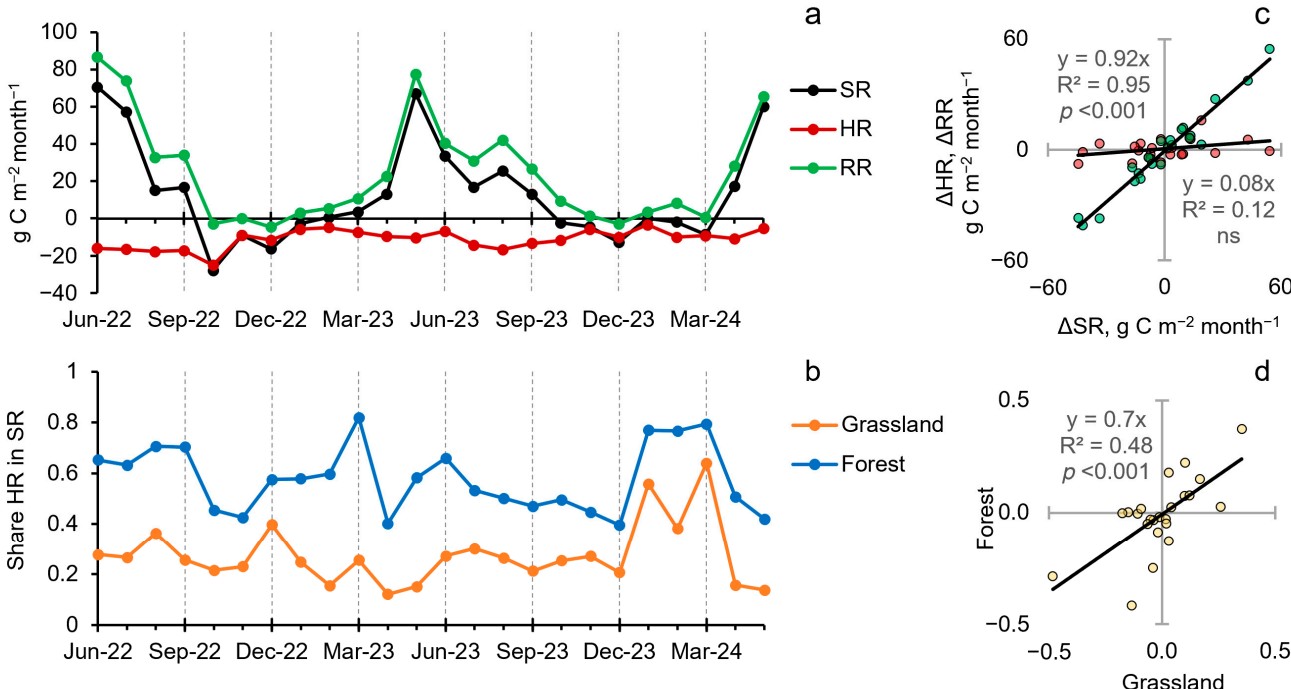

**Figure 6.** Differences in the total monthly soil (SR), SOM-derived heterotrophs (HR), and root-derived (RR) respiration between the grassland and forest ecosystems (**a**); a positive value means more intensive fluxes in the grassland. Dynamics of the share of monthly HR in SR values in the ecosystems (**b**). The relationships between increments of HR or RR and SR (**c**) values presented in (**a**), as well as the increments of HR shares between the forest and the grassland (**d**).

Surprisingly, there were no clear seasonal dynamics of the shares of HR and RR in the total monthly SR in contrast to the differences between ecosystems (Figure 6b). However, a small yet significant relationship was found between the dynamics of HR contributions in the two ecosystems studied (Figure 6d). The annual shares of HR were 57–67% in the forest and 28% in the grassland with minor variation across seasons. In the grassland, the minimum of HR share was in the spring (17–19%) and the maximum in the summer (32–34%), comprising 26–37% during the autumn and winter periods. In the forest ecosystems, the maximum share of HR was registered in the summer (62–76%) and comprised 52–62% during all other seasons. Nevertheless, the changes in HR shares between the two years differed between the two ecosystems. In the grassland, the contribution of HR during the cold season increased from 27% in the first year to 40% in the second year. However, this dynamic did not lead to a change in the annual share of HR. In the forest, the share of HR during the warm season decreased year-to-year from 69% to 57%, which led to a decrease in the annual share of HR from 67% to 57%.

The total annual SR flux was 1.3 times higher in the grassland compared to the forest for both years studied (Table 3). These differences were formed during a warm period (1.4–1.5 times higher), and especially during spring (1.6–1.9 times higher) when rapid RR growth was observed in the grassland. During the cold season, the alternate total SR was 1.2−1.3 times higher in the forest compared to grassland, due to the differences in HR values between the ecosystems.

Contributions of total winter SR, HR, and RR to annual fluxes were similar and comprised 10–15% in the forest ecosystem and 5–13% in the grassland. The cold season (November–April) contributed to the total annual SR, HR, and RR fluxes of 18–26% and 11–23% in the forest and grassland ecosystems, respectively.

**Table 3.** Mean values of the seasonal total soil (SR), SOM-derived heterotrophs (HR) and root-derived (RR) respiration ($CO_2$) in the forest (F) and the grassland (G) ecosystems for two years observation.

| Period [1] | SR | | | | HR | | | | RR | | | |
|---|---|---|---|---|---|---|---|---|---|---|---|---|
| | 2022–2023 | | 2023–2024 | | 2022–2023 | | 2023–2024 | | 2022–2023 | | 2023–2024 | |
| | F | G | F | G | F | G | F | G | F | G | F | G |
| Summer (January–August) | 214 (10) | 333 (13) | 194 (9) | 262 (7) | 163 (7) | 112 (5) | 120 (2) | 83 (9) | 51 (13) | 222 (14) | 74 (9) | 179 (11) |
| Autumn (September–November) | 141 (7) | 127 (5) | 118 (6) | 122 (5) | 83 (3) | 33 (2) | 64 (2) | 33 (3) | 59 (7) | 94 (6) | 54 (7) | 90 (6) |
| Winter (December–February) | 55 (5) | 37 (4) | 72 (5) | 63 (3) | 32 (3) | 10 (2) | 43 (3) | 23 (4) | 23 (6) | 27 (4) | 28 (6) | 40 (5) |
| Spring (March–May) | 91 (5) | 170 (8) | 106 (5) | 174 (12) | 56 (3) | 29 (2) | 55 (3) | 33 (3) | 35 (6) | 142 (8) | 51 (6) | 141 (12) |
| Cold season (November–April) | 101 (6) | 78 (5) | 123 (6) | 101 (3) | 59 (4) | 21 (2) | 74 (3) | 40 (5) | 43 (7) | 58 (5) | 49 (7) | 61 (6) |
| Warm season (May–October) | 402 (16) | 587 (16) | 370 (11) | 525 (15) | 277 (8) | 164 (6) | 210 (4) | 134 (9) | 125 (15) | 424 (17) | 160 (12) | 391 (17) |
| Year | 504 (15) | 667 (16) | 495 (13) | 626 (15) | 336 (9) | 185 (6) | 284 (5) | 174 (11) | 168 (16) | 483 (18) | 210 (14) | 453 (18) |

[1] June was taken at the beginning of the year. Confidence intervals (95%) are shown in brackets.

## 4. Discussion

### 4.1. Vegetation Type-Driven Effects on $CO_2$ Fluxes

We did not observe marked differences between the soils studied, excluding some distinctions in the topsoil layer (Table 1). The soil in the forest had a slightly higher pH in the 0–5 cm layer, as well as a higher $C_{org}$ content in the 0–10 cm layer in comparison to grassland soil. A larger amount of $C_{org}$ in the top 10 cm of the forest soil against grassland is common for sandy soils [33]. In contrast to the differences in $C_{org}$ and $N_{tot}$ content between forest and grassland soils, the C/N ratio in forest soil was systematically higher at all depths. Nevertheless, there was no observed effect of $C_{org}$ and $N_{tot}$ content on the $C_{mic}$. Since overall soil properties varied slightly, we assume that the main differences in soil $CO_2$ fluxes between these ecosystems are primarily due to the influence of plant communities [34], rather than being driven by differences in soil properties. However, higher soil organic carbon and C/N ratio in the forest can result in higher basal respiration of the soil, ceteris paribus.

One of the substantial differences between the ecosystems is the existence of the litter horizon in the forest soil and the sod horizon in the grassland soil. This reflects the predominant pathways of carbon entering the soil, its transformation, and release [35,36]. Additionally, a relative abundance of active thin roots in the grassland provides differences in an SR structure and its dynamics. The second major factor affecting the SR flux dynamics is the difference in the microclimate. Because of strong isolation of soil by vegetation canopy and litter layer from direct solar radiation, the forest soil was cooler and wetter than the soil in the grassland. The low water-holding capacity of the sandy soil also contributes to these differences. Thus, the consequences of weather anomalies on the hydrothermal regime of the soil in the grassland may be stronger than in the forest [37]. The microclimate effect can mediate the response of communities to extreme weather conditions [38] as well as the $CO_2$ source ratio in the soil.

### 4.2. Structure of Soil $CO_2$ Efflux

One of the main reasons for the lack of HR and RR data from regular long-term observations is the difficulty and complexity of separating the components of soil respiration in the field [19,20]. Most of the approaches are not suitable for long-term, full-year, frequent (not less than 1 per month, but, actually, 1/Mo is not frequent) observations. This is probably why the few regular observations of HR in the field are based on simple methods for separating HR from SR, such as girdling, trenching, root exclusion, and component integrations [39,40]. However, these methods vary in the degree to which they isolate HR.

For instance, the HR contribution estimates we obtained here were lower than the previously conducted SR component determined in these areas using the component integration method: 45–80% vs. 80–99% in the forest and 15–40% vs. 40–90% in the grassland [22,41]. These differences were expected due to the methodological underestimation of HR from the rhizosphere and root decomposition [42,43]. If the RR contribution obtained using the component integration method is subtracted from our RR estimates, the contribution of the rhizosphere and detritus to SR will be ½ in the grassland and ⅓ in the forest. On the other hand, these will contribute to a total HR of ⅔ in the grassland and ⅖ in the forest. This is a very rough estimate, but it provides an overview of the sources of $CO_2$ emissions in the two contrasting ecosystems.

The absence of clear seasonal patterns in HR share was unexpected. This means that $CO_2$ fluxes from roots, rhizodeposits, and detritus vs. from SOM and forest floor litter are generally in balance throughout the year. However, the mean share of HR in total seasonal SR during the offseason was a little, but significantly, lower than during summer or winter. These mean differences (in absolute share HR in SR) were 7% in the forest and 10% in the grassland (two-way ANOVA, $p < 0.05$). The composition of soil microbial communities changes with the seasons [44]. A possible explanation for the decline of HR share in SR is that microbial communities that consume SOM might have a longer adapting period to changes in the temperature regime of the soil, which is characterized by warm and cold seasons. Another explanation for this phenomenon could be the death and decay of the thin roots, which are absent in the HR chambers.

Apparently, the source of the seasonal variation in HR, which was previously found for the studied area [22,41], is associated with short-lived soil carbon pools, such as rhizodeposits, root, and detritus/litter. If these pools reflect the seasonal dynamics of the function of the ecosystems, seasonal SR differences between the two ecosystems will be determined entirely by the dynamics of these short-lived components of the soil carbon cycle, as we assumed in the first hypothesis. Indirectly, this can be seen by the close relationship between the HR dynamics in the grassland and the forest ($R^2 = 0.86$, n = 105) and the explanation of the differences in SR between ecosystems, which is almost exclusively related to RR dynamics ($R^2 = 0.96$, n = 24). Additionally, it is difficult to expect large differences in the SOM mineralization rates of soils with close properties without direct or indirect influence of ecosystem communities. Therefore, systematic monitoring in two or more contrasting ecosystems will allow us to extract additional information about the carbon cycle processes under various weather conditions.

### 4.3. Soil Respiration and $CO_2$ Sources Throughout Climate Change

The definition of drought can affect the conclusions drawn from manipulative experiments [45] and monitoring. Two years with similar mean annual temperatures and total precipitation amounts have allowed us to study several patterns in precipitation distribution. During summer, decreases in total monthly SR and HR were related both to the deficit of the total amount of precipitations and to a decrease in stratiform precipitations in favor of convective precipitations. Soils do not retain more water than their water-holding capacity. Thus, a portion of intense precipitation is excluded from biological consumption due to surface runoff and subsurface drainage. Rising precipitation in the warming world is a fact, but it has complex patterns [46]. Seasonal redistribution of precipitation can increase the length of the dry period, as well as a shift in the total amount of precipitation between seasons [47–49]. These are examples of an increase in aridity in some regions, despite the absence of a decrease in the total precipitation amount. There were specific effects of dry periods of varying lengths and intensities on HR and SR [50,51]. The link between $CO_2$ efflux reduction and the duration of the dry period is consistent with our previous results from the field experiment [50,52]. The magnitude and direction of the effects can be determined by comparing the positive impact of temperature on biochemical reactions with the negative impact of decreased soil moisture on substrate transport and availability [50]. Thus, soil hydrology properties will establish a threshold of the duration

of dry periods and a total amount of precipitation for effect on SR and HR. The sandy soil in the research area has a low water-holding capacity, making the ecosystems more vulnerable to dry periods. Previously, contrary trends in long-term SR dynamics were observed between this forest and a forest placed on the opposite side of the Oka River [53]. Perhaps these different trends in SR reflect the contrasting soil textures of the two forest ecosystems (sandy vs. loamy) and their respective water retention abilities. Therefore, it is necessary to account for not only the indicators of overall moisture availability, but also parameters that reflect the distribution of precipitation over time, such as the duration of dry periods and land features.

Rather warm and wet October 2022 favored fresh litter decomposition in the forest. This will not only affect the actual amount of $CO_2$ emissions in different years, but it may also have an impact on the microbial immobilization of carbon and nitrogen from labile organic compounds during the autumn–winter period [54]. In a broader sense, the effects relate to those that are a legacy from the conditions of ecosystem functioning in previous periods [55]. We observed a consistent 6% decrease in annual HR and RR from the first to the second year in the grassland, attributed to a dry summer at the beginning of our observation. The aforementioned effect of an increased duration of dry periods in the second year may include both the direct impact of weather conditions and the legacy effect of a deficit in summer precipitation from the previous year. In contrast, this has not been observed in the forest, where the topsoil moisture content was higher than in the grassland. In the forest, the decrease in HR was balanced by the increase in RR. Therefore, weather conditions affected the contribution of $CO_2$ sources to SR, but not its total amount in the forest. These observations, coupled with the specific response in HR and RR to the dry periods, are consistent with the second hypothesis. It is important in the context of specific responses of grassland and forest ecosystems to anomalous weather conditions [38,56].

Long-term observations of SR in these ecosystems reveal their negative trends in total annual SR fluxes [57,58]. They are accompanied by an increase in the annual air temperature, aridity during a growing season, and a reduction of a period with stable snow cover [53,57]. However, the contribution of certain soil $CO_2$ sources to the observed decrease in SR is not known. This is crucial to estimate carbon uptake by ecosystems and the risk of carbon loss due to climate extremes. For example, the proposed acceleration of the turnover of carbon pools under elevated $CO_2$ levels in the atmosphere [59] could lead to an increased risk of losing more carbon during secondary succession due to the previous depletion of recalcitrant SOM. Therefore, long-term continuous monitoring of soil respiration sources is just as important as monitoring soil respiration itself and the ecosystem's $CO_2$ fluxes in general [60].

## 5. Conclusions

The main differences in seasonal dynamics of $CO_2$ efflux from soil (SR) between nearby grassland and forest ecosystems were related to root-associated $CO_2$ (root with rhizosphere microorganisms; RR). This component of SR was also more valuable for the impact of prolonged dry periods than the respiration of soil heterotrophs (SOM-derived and leaf litter; HR). Remarkable drying of grassland soil results in the more pronounced SR and HR decreases during dry periods in the grassland compared to the forest one. In contrast to HR, the total seasonal RR fluxes were not consistent across the two ecosystems at the interannual scale. It remains to be investigated whether this is a direct result of the weather conditions or the dynamics of more complex ecosystem processes. The absence of pronounced seasonal variation of HR contribution to SR (primarily SOM-derived) is likely due to the fact that the main source of seasonal variability in HR arises from short-lived components of the soil C cycle, such as rhizodeposits, dead fine roots, detritus, and litter. Consequently, developing long-term monitoring programs that include the assessment of soil respiration components, particularly short-lived ones, would strongly reduce uncertainty in carbon balance estimates for terrestrial ecosystems.

**Author Contributions:** Conceptualization, D.K., I.K. and V.L.d.G.; methodology, D.K. and V.L.d.G.; software, A.S.A.; validation, S.K., A.S.A. and Y.K.; formal analysis, A.S.A.; investigation, D.K.; resources, I.K.; data curation, D.K., V.L.d.G. and D.S.; writing—original draft preparation, D.K.; writing—review and editing, D.K., I.K. and Y.K.; visualization, S.K. and D.S.; supervision, project administration and funding acquisition I.K. All authors have read and agreed to the published version of the manuscript.

**Funding:** Data of heterotrophic soil respiration, analysis, and preparation of the manuscript were carried out within the framework of the state assignment of the Ministry of Science and Higher Education of the Russian Federation (No. 122111000095-8). Data on soil respiration were obtained within the framework of the state assignment of the Ministry of Science and Higher Education of the Russian Federation (project No. 122040500037-6).

**Data Availability Statement:** The data presented in this study are available upon reasonable request from the authors.

**Acknowledgments:** We give thanks to Vera Ableeva (Station of Background Monitoring, Roshydromet), who provides us the meteorological data set for the study area. We are grateful to the Prioksko-Terrasny Nature Biosphere Reserve for the opportunity to conduct monitoring and for the organizational support provided for our research, and the RUDN University Strategic Academic Leadership Program. The authors extend their appreciation to the Deanship of Scientific Research at Princess Nourah bint Abdulrahman University for funding this work through the Visiting Researcher Program.

**Conflicts of Interest:** The authors declare no conflicts of interest. The funders had no role in the design of the study; in the collection, analyses, or interpretation of data; in the writing of the manuscript; or in the decision to publish the results.

## Appendix A

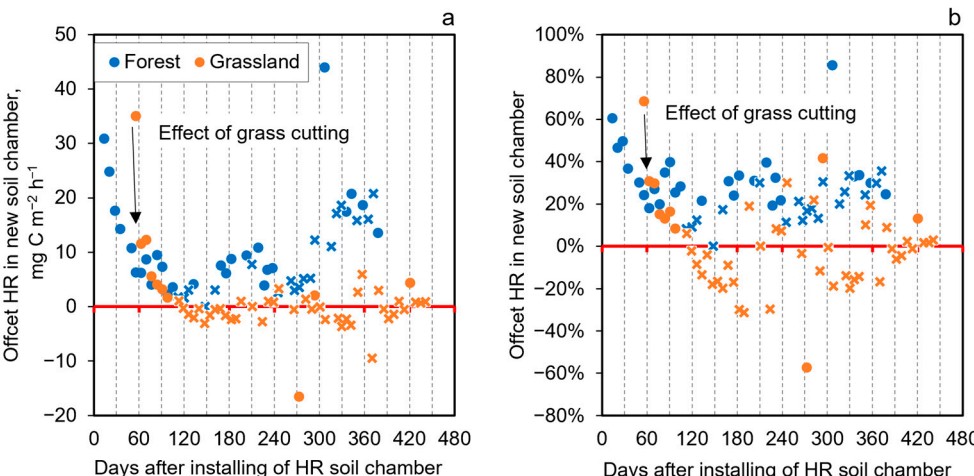

**Figure A1.** Offset of SOM-derived respiration of heterotrophs (HR) measured in young (installed in June–July 2023) soil chambers relative to mature (installed in May 2022) soil chambers, in terms of absolute (**a**) and relative (**b**) values. A cross mark indicates days with no significant differences (*t*-test with equal variances, n = 4–5, $p < 0.05$).

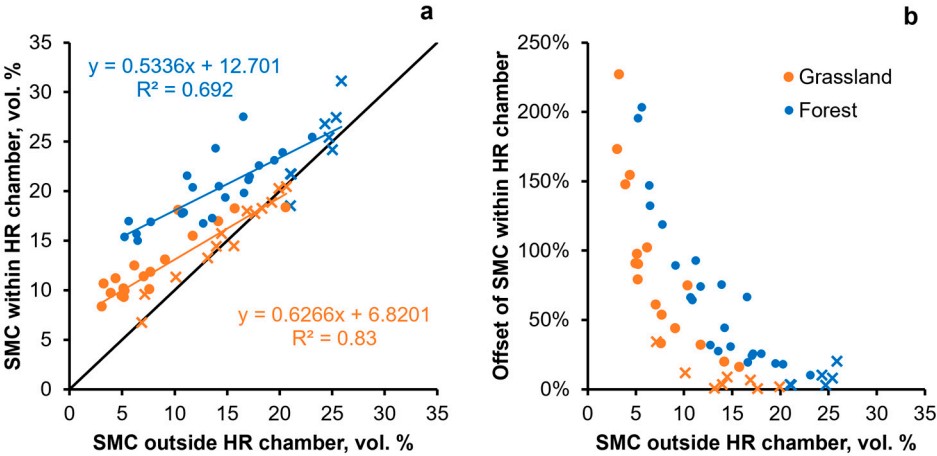

**Figure A2.** Offset of soil moisture content (SMC) at the depth of 0–6 cm within soil chambers without roots relative to surrounding intact soil, in terms of absolute (**a**) and relative (**b**) values. A cross mark indicates days with no significant differences (*t*-test with equal variances, n = 5–10, *p* <0.05).

## Appendix B

**Table A1.** Summary statistics of soil respiration (SR) and respiration of SOM-derived microorganisms (HR) in the forest and grassland ecosystems for the two consequent years (June was taken at the beginning of the year) from 2022 to 2024 (n = 46–48; mg C m$^{-2}$ h$^{-1}$).

| Flux | Ecosystem | Year | Min | Q1 | Median | Q3 | Max |
|------|-----------|------|-----|-----|--------|-----|-----|
| SR | Forest | 2022–2023 | 13.9 | 38.7 | 51.8 | 72.4 | 229.4 |
| | | 2023–2024 | 11.4 | 36.0 | 53.3 | 91.0 | 125.9 |
| | Grassland | 2022–2023 | 13.1 | 27.7 | 65.5 | 105.7 | 239.9 |
| | | 2023–2024 | 12.2 | 32.7 | 65.5 | 124.3 | 227.1 |
| HR | Forest | 2022–2023 | 9.5 | 19.7 | 32.4 | 53.5 | 102.5 |
| | | 2023–2024 | 8.3 | 21.6 | 27.1 | 51.2 | 80.3 |
| | Grassland | 2022–2023 | 2.3 | 7.9 | 16.9 | 27.5 | 71.0 |
| | | 2023–2024 | 2.9 | 9.7 | 15.7 | 30.6 | 56.3 |

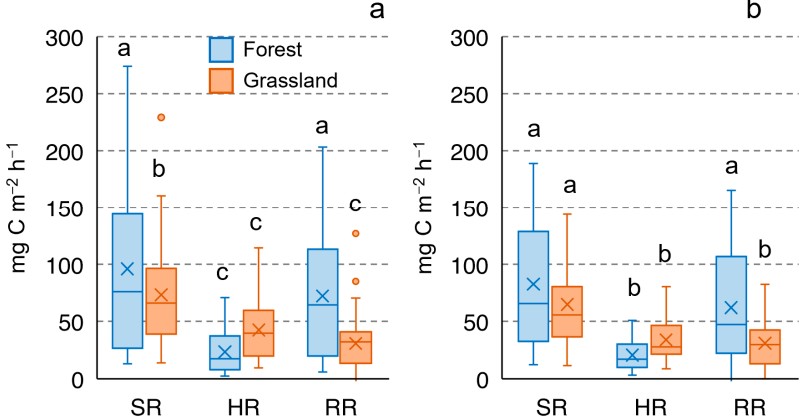

**Figure A3.** Respiration of soil (SR) and its SOM-derived (HR) and root-derived components (RR) during 2022–2023 (**a**) and during 2023–2024 (**b**): the mean (cross), the median (bar), lower (Q1), and upper (Q3) quartiles ("boxes"); X1 = Q1 − 1.5 IQR (interquartile range, IQR = Q3 − Q1) and X2 = Q3 − 1.5 IQR ("moustaches"); all data are shown as dots. Different letters indicate pairs of average values, the differences of which are detected during the multiple comparison procedure (Tukey test, $\alpha$ = 5%) after two-way ANOVA (Flux component × Ecosystem).

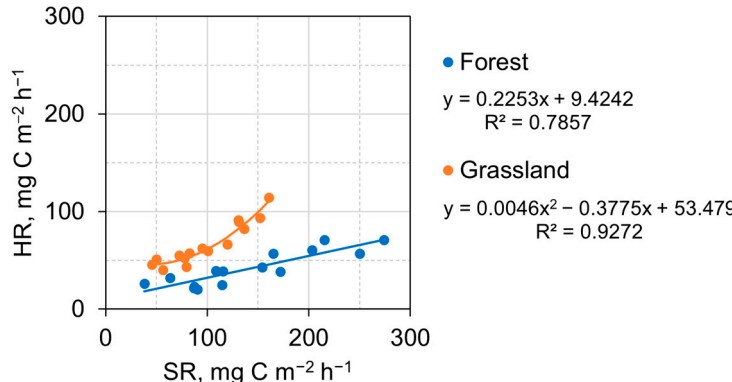

**Figure A4.** The regression functions of SOM-derived microorganisms respiration rate (HR) based on soil respiration rate (SR) was developed using data from 20 July 2022 to 29 September 2022, which was then used to reconstruct HR for the period from 1 July 2022 to 14 July 2022.

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
