# Peer review of "Vegetation and Precipitation Patterns Define Annual Dynamics of CO2 Efflux from Soil and Its Components"

_land, doi:10.3390/land13122152_

Round 1
Reviewer 1 Report
Comments and Suggestions for Authors
1 - The study is well designed
2 - The methodology is appropriate and well documented
3 - The results contribute to our understanding of soil respiration and carbon sequestration
4 - The English is excellent, I did notice a couple of places where the choice of phrasing looks like a translation from another language, like lines 212 to 213 discussing snow melt
Author Response
The authors are grateful to the reviewer for their efforts and positive assessment of the work.
Comments: The English is excellent, I did notice a couple of places where the choice of phrasing looks like a translation from another language, like lines 212 to 213 discussing snow melt
Response: We agree that sentence construction is complex. Therefore, we have simplified it while maintaining context.
[Before] A 50-year record amount of precipitation fell in December. However, it did not lead to a sharp increase in a snow cover amount due to some of them falling in liquid during thaws.
[After, changed parts highlighting red] A 50-year record amount of precipitation (liquid equivalent, mm) fell in December. However, these precipitations did not lead to a sharp increase in a snow cover amount due to thaws.
We also reviewed the text again and made efforts to improve it by simplifying complex structures. Changes in the document have been highlighted in blue.
Reviewer 2 Report
Comments and Suggestions for Authors
Long-term monitoring of soil heterotrophic respiration (HR) and autotrophic respiration (RR) is very significant for exploring soil carbon cycle and estimating the global carbon budget. Through measuring the seasonal dynamics of HR and SR in forest and grassland ecosystems for 2 consecutive years, the authors found the SR was higher in the grassland than in the forest, but the HR was lower in the grassland than in the forest, and the HR component of CO2 efflux from soil averaged for 62% in the forest and 28% in the grassland. Despite the interest brought by this manuscript, there are several concerns that need to be addressed prior to its publication.
Abstract
Lines 28-29: The existence of this sentence is meaningless. It lacks favorable evidence and is also not an important finding of this paper.
Lines 30-31: Individual keywords are unnecessary, such as summer drought.
Introduction
This part emphasizes the importance of long-term soil respiration monitoring and the shortcomings of HR monitoring, which lacks a good grasp of the topic. I suggest that the authors strengthen the relevant introduction of the effects of different ecosystems on soil respiration and heterotrophic respiration, including the relationship between heterotrophic respiration and total respiration and the possible causes.
Materials and Methods
Line 133:Table 1. We would like to see the statistical results of the ANOVA, so as to conveniently correspond to the results in lines 313-318. The pH, Corg and C/N ratio in 0-10 layer in the forest should be significantly higher than in the grassland. This does not correspond well with the results in the discussion part.
Lines 132-175: I did not find a description of repetition (sample size n) in this section, please add a relevant description.
Results
For some results, I suggest using ANOVA (Figure 2) or repeated measure ANOVA (Figure 3, 4) model for analysis. The author attempts to explore the effects of climate change such as drought on soil respiration. So, I propose to add data fragments about drought stages in this section for relevant analysis.
Discussion
I suggest that the author focus on the reasons for the core conclusions. For example, Lines 332-345, this paragraph emphasis on the importance of long-term continuous monitoring of SR or HR, but is not very relevant to the core results of the article.
Line 313-318: These conclusions are rather arbitrary and suggest a more accurate description and inference.
Conclusions
This part is somewhat redundant, and there are some unnecessary subjective content (lines 462-467), it is recommended to summarize the main results.
Comments on the Quality of English LanguageEnglish expression needs to be improved.
Author Response
We appreciate the reviewer for their constructive comments, which have contributed to the improvement of the article. Below, we have provided our responses to the comments. All changes in the revised paper have been highlighted in blue.
Comments 1: Lines 28-29: The existence of this sentence is meaningless. It lacks favorable evidence and is also not an important finding of this paper.
Response 1: We agree. This sentence was removed from the abstract in the revised paper. We also added some values to the abstract to illustrate the results.
Comments 2: Lines 30-31: Individual keywords are unnecessary, such as summer drought.
Response 2: We have revision keywords. "climate change" and "summer droughts" were removed.
Comments 3: This part emphasizes the importance of long-term soil respiration monitoring and the shortcomings of HR monitoring, which lacks a good grasp of the topic. I suggest that the authors strengthen the relevant introduction of the effects of different ecosystems on soil respiration and heterotrophic respiration, including the relationship between heterotrophic respiration and total respiration and the possible causes.
Response 3: We removed the section discussing the difficulties associated with separating soil respiration and instead focused on ecosystems and the components of soil respiration.
[Added text] The type of ecosystem and a vegetation cover have a strong effect on SR because they are responsible for the input of organic compounds into soil, microclimate, biodiversity and food webs, as well as soil properties. Reflecting the total biological activity of an ecosystem, SR and, particularly HR are controlled both by the climatic conditions and vegetation type [16–18]. The average contribution of root respiration to the total SR flux from the soils varies from 38% in agrocenoses to 63–72% in tundra and northern taiga ecosystems [1]. However, the contribution of RR to SR can vary significantly depending on the time, place, and method of CO2 flow separation, with values ranging from less than 10% to more than 90% [19,20]. The seasonal and interannual soil CO2 fluxes and its structure linked with the ecosystem type [21–23], whereas the reaction the SR to the climatic changes and extremal events [24] may be associated with the individual soil carbon pools with different turnover times [25]. The importance of regularly monitoring the SR structure is highlighted by the presence of seasonal dynamics specific to different ecosystems [22,26,27]. The heterogeneous response of separate SOM components to changing climatic conditions deepens our understanding of the relationship between global warming and the terrestrial carbon cycle.
Comments 4: Line 133:Table 1. We would like to see the statistical results of the ANOVA, so as to conveniently correspond to the results in lines 313-318. The pH, Corg and C/N ratio in 0-10 layer in the forest should be significantly higher than in the grassland. This does not correspond well with the results in the discussion part.
Response 4: We included more detailed results for soil properties at depths of 0-5, 5-10, 10-20, and 20-30 cm. We also conducted a two-way ANOVA to compare the two ecosystems across different soil depths. The results have been added to Table 1. Additionally, we revised the discussion in lines 313-318. More details in the response to Comment 8.
Comments 5: Lines 132-175: I did not find a description of repetition (sample size n) in this section, please add a relevant description.
Responded 5: Thank! We added this information. N = 5.
Comments 6: For some results, I suggest using ANOVA (Figure 2) or repeated measure ANOVA (Figure 3, 4) model for analysis. The author attempts to explore the effects of climate change such as drought on soil respiration. So, I propose to add data fragments about drought stages in this section for relevant analysis.
Responded 6: We calculated two-way ANOVA + Tukey test for results on the Figure 2 (we moved it to Appendix B1 after revision). We, also, added two figures for illustrate effect of prolonged dry periods to the CO2 fluxes from soil (Figure 3 and 4 in revision paper).
[Added text with Figure 3 and 4] The remarkable decline of SR rate was observed during the dry periods in August 2022, June 2023 and September 2023 (Figure 3). Soil moisture content was a substantial factor influencing SR dynamics during these periods (Figure 4). We estimated the effect size of a dry period as the difference between CO2 fluxes on the first day of the dry period and the minimum CO2 fluxes measured during that period. The decrease in CO2 efflux during August 2022 was more pronounced compared to June 2023 and September 2023, due to a longer dry period (37 days vs. 21-22 days, respectively). The decline of SR and HR in the grassland during the dry periods was more than in the forest: 56% vs 39% across all fluxes and periods (paired t-test, n = 6, p <0.001). There was found HR were less affected by dry periods compared to SR and, consequently, RR (Figure 3).
We calculated two-way ANOVA for Figure 5 (figure number from revision paper)
[Added text for Figure 5] For the formal analysis, we divided the year into two seasons: October to March and April to September. Two-way ANOVA indicated a likely interaction between ecosystem type and the seasonal factor for SR (p = 0.006), along with a formally non-significant effect of the ecosystem at the annual scale (p = 0.06). For HR, the analysis revealed no interaction between the ecosystem and the season (p = 0.59), while a significant ecosystem effect was observed (p = 0.0006). All possible combinations of ecosystem and seasonal factors were significant for RR (p << 0.001).
We added correlation between increments of HR/RR and SR, as well as the increments of HR shares between forest and grassland in the Figure 6 (figure number from revision paper) to strengthen the results regarding the contribution of SR components to the seasonal dynamics of SR differences between the ecosystems.
Comments 7: I suggest that the author focus on the reasons for the core conclusions. For example, Lines 332-345, this paragraph emphasis on the importance of long-term continuous monitoring of SR or HR, but is not very relevant to the core results of the article.
Responded 7: We removed text (lines 332-345 in the previous edition). We also changes the section name for 4.2.
[Before] From observational data to soil organic matter pools and functions.
[After] Structure of soil CO2 efflux.
We also integrated the results from the present study with those obtained earlier by us in a field experiment.
[Added text] The link between CO2 efflux reduction and the duration of the dry period is consistent with our previous results from the field experiment [50,52].
Some corrections were made to improve the text.
Comments 8: Line 313-318: These conclusions are rather arbitrary and suggest a more accurate description and inference.
Responded 8: We improved the section to be more accurate and in agreement with the results in Table 1.
[Before] The close proximity of the two contrasting ecosystems has led to similar properties in the soils. A slightly larger amount of Corg in the top 10 cm of the forest soil against grassland is common for sandy soils [29], reflected in the larger soil C/N ratio under the forest, but without effects on the Cmic content. Therefore, the main differences in soil CO2 fluxes from these ecosystems as a whole are due to the influence of plant communities [30], rather than being driven by differences in soil properties.
[After] We did not observe marked differences between the soils studied excluding some distinguishes in the topsoil layer (Table 1). The soil in the forest had a slightly higher pH in the 0–5 cm layer, as well as a higher Corg content in the 0–10 cm layer in comparison to grassland soil. A some larger amount of Corg in the top 10 cm of the forest soil against grassland is common for sandy soils [33]. In contrast to the differences in Corg and Ntot content between forest and grassland soils, the C/N ratio in forest soil was systematically higher at all depths. Nevertheless, there was no observed effect of Corg and Ntot content on the Cmic. Since overall soil properties varied slightly, we assume that the main differences in soil CO2 fluxes between these ecosystems are primarily due to the influence of plant communities [34], rather than being driven by differences in soil properties. However, higher soil organic carbon and C/N ratio in the forest can result in higher basal respiration of the soil, ceteris paribus.
Comments 9: This part is somewhat redundant, and there are some unnecessary subjective content (lines 462-467), it is recommended to summarize the main results.
Responded 9: We rewrote this section, taking into account the reviewer's recommendations.
[Before] Two years monitoring of CO2 efflux from soils (SR) with its partitioning for contribution of soil heterotrophs (SOM-derived + leaf litter; HR) and root-associated CO2 (root with rhizosphere microorganisms; RR) were provided at nearby locations with similar soils but contrast vegetation cover: forest vs grassland. It allowed to estimate not only the contribution of specific components of SR but also understand the influence of weather conditions. The main distinction in SR values between the forest and grassland was related to the share of RR. The dynamics of root component explained most of the differences in temporal variation of SR values between these two ecosystems.
An increase in aridity can be associated not only with a lack of precipitation but also with a longer duration of dry periods, even when the climatic average precipitation amount remains unchanged. The response of the RR to an increase in aridity determines the interannual variability of the SR, especially in the grassland. However, it remains to be investigated whether this is a direct result of the dry periods themselves or a cumulative effect of the previous dry summer.
The absence of pronounced seasonal variation of HR contribution to SR (primarily SOM-derived) is likely due to the fact that the main source of seasonal variability in HR arises from short-lived components of the soil C cycle, such as rhizodeposits, died roots, detritus, and litter. In this instance, the accumulation and decomposition of these components will determine the differences in the amount and direction of carbon flux between ecosystems and the atmosphere over time. Consequently, developing long-term monitoring programs including the measurements of soil respiration components would strongly reduce uncertainty in carbon balance estimates in terrestrial ecosystems.
[After] The main differences in seasonal dynamics of CO2 efflux from soil (SR) between nearby located grassland and forest ecosystems were provided by root-associated CO2 (root with rhizosphere microorganisms; RR). This component of SR was also more valuable for impact of prolonged dry periods than respiration of soil heterotrophs (SOM-derived + leaf litter; HR). Remarkable drying of grassland soil results in the more pronounced SR and HR decrease during dry periods in the grassland compared to the forest one. In contrast to HR, the total seasonal RR fluxes were not consistent across the two ecosystems at the interannual scale. It remains to be investigated whether this is a direct result of the weather conditions or the dynamics of more complex ecosystem processes. The absence of pronounced seasonal variation of HR contribution to SR (primarily SOM-derived) is likely due to the fact that the main source of seasonal variability in HR arises from short-lived components of the soil C cycle, such as rhizodeposits, died roots, detritus, and litter. Consequently, developing long-term monitoring programs that include the assessment of soil respiration components, particularly short-lived ones, would strongly reduce uncertainty in carbon balance estimates for terrestrial ecosystems.
Reviewer 3 Report
Comments and Suggestions for Authors
The paper brings interesting contributions about estimating the seasonal dynamics of SOM-derived and root-derived components of SR in forest and grassland ecosystems.
The title is adequate and representative of what the article presents and the abstract is in general according with the entire article.
In the introduction there are presented the two objective of the article and citations are current.
The research locations and the methodology are distinctly described, with sufficient details to allow reproduction of experiments.
The processed data and the statistics are explained correctly, with tables and figures corresponding to the data in the text.
The results follow the two objectives of the research and the article ends with the conclusions that point out the importance of the research results. All citations in the text can be found in the bibliography section and are consistent with the research carried out
In my opinion it doesn't exist plagiarism and also there is no conflict of interest.
Author Response
We appreciate the reviewer for their efforts and positive assessment of the work.
We also made some corrections to the text. Changes in the document have been highlighted in blue.